

# Model Enforced Post-Process Correction of Satellite Aerosol Retrievals

Antti Lipponen[1], Ville Kolehmainen[2], Pekka Kolmonen[1], Antti Kukkurainen[1], Tero Mielonen[1], Neus Sabater[1], Larisa Sogacheva[1], Timo H. Virtanen[1], and Antti Arola[1]

[1]Finnish Meteorological Institute, Atmospheric Research Centre of Eastern Finland, Kuopio, Finland
[2]Department of Applied Physics, University of Eastern Finland, Kuopio, Finland

**Correspondence:** Antti Lipponen (antti.lipponen@fmi.fi)

**Abstract.** Satellite-based aerosol retrievals provide a timely global view of atmospheric aerosol properties for air quality, atmospheric characterization, and correction of satellite data products and climate applications. Current aerosol data products based on satellite data, however, often have relatively large biases relative to accurate ground-based measurements and distinct levels of uncertainty associated with them. These biases and uncertainties are often caused by oversimplified assumptions and
approximations used in the retrieval algorithms due to unknown surface reflectance or fixed aerosol models. Moreover, the retrieval algorithms do not usually take advantage of all the possible observational data collected by the satellite instruments and may, for example, leave some spectral bands unused. The improvement and the re-processing of the past and current operational satellite data retrieval algorithms would become a tedious and computationally expensive task. To overcome this burden, we have developed a model enforced post-process correction approach that can be used to correct the existing and
operational satellite aerosol data products. Our approach combines the existing satellite aerosol retrievals and a post-processing step carried out with a machine learning based correction model for the approximation error in the retrieval. The developed approach allows for the utilization of auxiliary data sources, such as meteorological information, or additional observations such as spectral bands unused by the original retrieval algorithm. The post-process correction model can learn to correct for the biases and uncertainties in the original retrieval algorithms. As the correction is carried out as a post-processing step, it allows
for computationally efficient re-processing of existing satellite aerosol datasets with no need to fully reprocess the much larger original radiance data. We demonstrate with over land aerosol optical depth (AOD) and Angstrom exponent (AE) data from the Moderate Imaging Spectroradiometer (MODIS) of Aqua satellite that our approach can significantly improve the accuracy of the satellite aerosol data products and reduce the associated uncertainties. We also give recommendations for the validation of satellite data products that are constructed using machine learning based models.

# 1 Introduction

Climate change is one of the most serious problems humankind is facing today. Despite the long and active research, the future climate projections still contain significant uncertainties, and anthropogenic aerosol forcing comprises currently the largest source of this uncertainty (Pachauri et al., 2014). A more accurate information about the aerosol optical depth (AOD)



and Angstrom exponent (AE) would help us improve our understanding of anthropogenic aerosol forcing and thus lead to a

significant reduction of the uncertainties in future climate projections. Another major global problem is air quality. In year 2017, 2–25% of all deaths globally were attributable to ambient particulate matter pollution (GBD 2017 Risk Factor Collaborators et al., 2018). To better monitor and understand air quality and pollution sources, near real-time global observations of aerosols are needed. In this respect, the only way to get wide coverage and near real-time information about atmospheric aerosols is to use satellite aerosol retrievals.

Satellite aerosol retrieval algorithms retrieve the aerosol optical properties such as AOD and AE given the satellite observed top-of-atmosphere radiances or reflectances and the measurement geometry information. Currently, satellite retrieval algorithms for multiple satellite instruments have been developed and the satellite aerosol data records span timeseries that are over 40 years long (Sogacheva et al., 2020). One of the most widely used satellite aerosol data products is based on the Moderate Imaging Spectroradiometer (MODIS) data (Salomonson et al., 1989) and the Dark Target algorithm (Levy et al., 2013). The

MODIS Dark Target data starts from year 2000 and global data is available from two satellites: Terra and Aqua. The expected error (EE) envelope for AOD in Dark Target data over land is estimated to be $\pm(0.05 + 15\%)$ resulting to relatively large uncertainties especially in regions with relatively low AOD. For more information about the concept of the EE envelope see, for example, Sayer et al. (2015).

To improve the existing aerosol data sets, machine learning based solutions have been used in many studies. Most of the

approaches that utilize machine learning employ a *fully learned* approach for the solution of a satellite retrieval. In the fully learned approach, a machine learning based model is trained to predict the values of the unknown aerosol parameters such as AOD given the measurement data (top-of-atmosphere radiances or reflectances) and observation geometry as the inputs. Neural network based fully learned aerosol retrievals are assimilated into NASA's MERRA-2 re-analysis model (Randles et al., 2017). In Di Noia et al. (2017), a fully learned AOD retrieval neural network model is used to retrieve the initial AOD for an iterative

retrieval algorithm. In Lary et al. (2009), a fully learned approach with MODIS retrieved AOD and the surface type as an additional inputs was used for the satellite AOD retrieval with MODIS data. The results of Lary et al. (2009) were evaluated using the accurate ground-based Aerosol Robotic Network (AERONET) data (Holben et al., 1998). With neural networks the authors were able to reduce the bias of the MODIS AOD data from 0.03 to 0.01, while with support vector machines even better improvement was reported - AOD bias was less than 0.001 and the correlation coefficient with AERONET was larger

than 0.99. In the above mentioned work, the validation was performed using all the available AERONET network stations both for training and validation. The split between the training and validation datasets was carried out using random splits of the pixels. With the random split of all pixels, the data samples from the same AERONET station were present both in training and evaluation datasets. This may lead to overly optimistic results as the model learns, for example, the surface properties at the locations of the AERONET stations and can thus predict the aerosol properties very accurately at these locations but may

not generalize the results to other regions very well. In Albayrak et al. (2013), a neural network based fully learned MODIS AOD retrieval model was trained and evaluated. In their model, in addition to MODIS reflectances and measurement geometry information, they used MODIS retrieved AOD and its quality flag as additional auxiliary inputs. The output of their model was AOD. They found their model to produce more accurate AOD retrievals than the operational MODIS Dark Target algorithm.





In Lanzaco et al. (2017), a slightly different type of machine learning based approach was used to improve satellite AOD
retrievals. The authors used MODIS AOD retrievals and local meteorology information as inputs to predict the AOD in South
America. This approach that combines the conventional AOD retrievals and local meteorology information was reported to
improve the AOD accuracy over the operational MODIS AOD. A problem in fully learned approaches is that they fully rely on
trained data and do not employ physics-based models in the retrievals. This may cause problems in the capability of the model
to generalize to cases in which the inputs are far outside the input space spanned by the training dataset.

Following the philosophy of a post-processing correction strategy, we have developed a new *model enforced* machine learn-
ing approach in which we exploit also the models and the physics-based satellite-derived aerosol retrieval product. More
specifically, we train the machine learning model for post-process correction of the approximation error in the result of the
conventional retrieval algorithm. While the post-process correction approach is new to satellite retrievals, it has been found to
perform better and produce more stable and accurate results than a fully learned approach in generation of surrogate simulation
models (Lipponen et al., 2013, 2018) and in medical imaging, where many of the inverse imaging problems are mathemati-
cally highly similar to the satellite retrieval problems, see for example Hamilton et al. (2019). The key advantages of the new
model enforced post-process correction approach are 1) the improved accuracy over the existing data products and existing
fully learned satellite data approaches, and 2) the possibility to post-process correct existing (past) satellite data products with
no need for full re-processing of the enormous satellite datasets. A reason why our approach outperforms the current state-
of-the-art fully learned machine learning retrievals is that the approximation error is a less complicated function for machine
learning regression than the full physics-based retrieval. In our proposed correction approach, we combine the best aspects of
the conventional retrievals and machine learning to get the full information content out of the satellite data. Our approach is
not limited to aerosols and is generally applicable to different types of satellite data products as long as suitable training data
for the model is available.

The manuscript is organized as follows. The proposed post-process correction of satellite aerosol retrievals is presented in
Section 2. In Section 3, the data and models used to test our approach are shown. Evaluation of the models is presented in
Section 4 and results are shown in Section 5. The conclusions are given in Section 6.

## 2 Post-process Correction of Satellite Aerosol Retrievals

Let $y$ be an accurate satellite aerosol retrieval (e.g. AOD or AE) so that

$$
\begin{aligned}
y &= f(\mathbf{x}) \\
&= \tilde{f}(\mathbf{x}) + \left[ f(\mathbf{x}) - \tilde{f}(\mathbf{x}) \right] \\
&= \tilde{f}(\mathbf{x}) + e(\mathbf{x})
\end{aligned}
\tag{1}
$$

where $f$ is an accurate retrieval algorithm and $\mathbf{x}$ contains all the algorithm inputs including the observation geometry and
satellite observations such as the top-of-atmosphere reflectances. An approximative retrieval algorithm is denoted by $\tilde{f}$. In
reality, due to uncertainties in the atmospheric properties and computational limitations among other reasons, it is not possible



to construct an accurate retrieval algorithm $f$ which is why an approximative algorithm $\tilde{f}$ is used instead. The discrepancy between the accurate and approximative algorithm retrievals, that is the approximation error corresponding to $\tilde{f}(\mathbf{x})$, is denoted by

$$e(\mathbf{x}) = f(\mathbf{x}) - \tilde{f}(\mathbf{x}). \tag{2}$$

To compute the corrected retrieval in the model enforced post-process correction of satellite aerosol retrievals, we use both the conventional retrieval algorithm $\tilde{f}$ and a machine learning based model $\hat{e}(\mathbf{x})$ to predict the realization of the approximation error $e(\mathbf{x})$, and Equation (1). Note that this is different from the fully learned model in which the aim is to emulate the accurate retrieval algorithm $f(\mathbf{x})$ with a machine learning model $\hat{f}(\mathbf{x})$. The approximation error $e(\mathbf{x})$ is typically less complicated function for machine learning regression than the full physics-based retrieval $f(\mathbf{x})$ thus resulting in more accurate and reliable

results with the model enforced correction than with a fully learned approach. For a chart of conventional retrieval, fully learned machine learning, and model enforced post-process correction approaches see Figure 1. Remark that as the training of the post process correction is based on existing satellite data and retrievals, the implementation can be done in a straightforward manner, for example, using black-box machine learning code packages. In addition, the post process correction model is also flexible with respect the choice of the statistical regression model, and the choice of the regression model can be tailored to different

retrieval problems separately.

## 3   Data and models

For testing the model enforced post-process correction, we use MODIS satellite aerosol retrieval data (AOD and AE) over land from Aqua satellite. Both the proposed model enforced correction model and a fully learned model as a reference are trained and tested using data from the ground-based AERONET measurements.

### 3.1   MODIS Dark Target

MODIS instruments are flying on board NASA's Terra and Aqua satellites. Terra was launched in year 1999 and the MODIS aerosol products currently span a relatively long time series of about 20 years. MODIS Dark Target aerosol data products are among the most widely used satellite aerosol data. In this study, we use the 10 km resolution MODIS Dark Target over land level 2 Collection 6.1 data of Aqua satellite (MYD04_L2) from years 2014–2018 (Levy et al., 2013). We use the AOD

retrievals at wavelengths 440, 550, and 660 nm to compute the AE with a least-squares linear fit in log-log-scale. In addition to aerosol quantities, we use the observation (satellite acquisition and illumination) geometry, land surface altitude, and retrieval quality flags as inputs for our models from the aerosol data products in our study.

### 3.2   AERONET

AERONET is a global network of sun photometers (Holben et al., 1998). AERONET has a Direct Sun data product which

contains both the AOD and Angstrom Exponent data that we will use in our study. AERONET data are most commonly used

as an independent data source for aerosol retrieval validation and all the data is publicly available at the AERONET website (http://aeronet.gsfc.nasa.gov/). An extensive description of the AERONET sites, procedures and data provided is available from this website. Ground-based sun photometers provide accurate measurements of AOD, because they directly observe the attenuation of the solar radiation without interference from land surface reflections. The AOD estimated uncertainty varies

spectrally from $\pm0.01$ to $\pm0.02$ with the highest error in the ultraviolet wavelengths (Eck et al., 1999). In this study, we will use the AERONET data for both model training and validation purposes. The AERONET data are divided in cross-validation to sets of training stations and validation stations for good generalization of the machine learning model.

### 3.3   Fully learned and model enforced post-process correction models for aerosol retrievals

In this study, in addition to the model enforced post-process correction model, we also train a fully learned model for the aerosol

retrieval to be used as a reference. We use the Random Forest (RF) regressor (Breiman, 2001) as our machine learning method to train all the machine learning based models. RF is an ensemble learning algorithm that uses regression trees as base learners. RFs can learn non-linear functions and they are relatively tolerant against overfitting. RFs have been shown to provide highly accurate results in many applications and they are relatively straightforward to train with a low number of hyperparameters to tune in the training. Training of the RFs can also be done with a relatively low computational costs and the trained models

are fast to evaluate. We use Python Scikit-Learn library implementation for the RFs (Pedregosa et al., 2011). During our work we also carried out preliminary tests with neural network based models trained with the same data as RFs but due to worse performance did not use them in the final evaluation.

Before training the final models we carried out a hyperparameter optimization for each of the models. In the hyperparameter optimization, we used an exhaustive 3D grid search and tested all possible combinations of hyperparameters in our candidate

sets using 2-fold cross validation with our training data. In the candidate set, we had three hyperparameters to be optimized:

- Number of trees: 100, 200, 400

- Maximum depth of a single tree: 30, 40, 50, 60

- Maximum number of features to consider when building the regression trees (as fraction of number of features): 100%, 80%, 60%, 40%

For other hyperparameters, the default values of the Scikit-Learn library were used. Based on the exhaustive grid search results, we averaged the hyperparameter values of the 10 best performing models measured with the explained variance metric. The hyperparameter values obtained by the averaging were used for the training of the final models. See derived values of the optimized hyperparameters in Table 1.

### 3.3.1   Fully learned model

The fully learned machine learning model $\hat{f}(\mathbf{x})$ takes the MODIS observation geometry information and the top-of-atmosphere reflectance information as inputs and directly predicts the AOD at 550 nm and AE. The input variables in the fully learned





models for both AOD and AE are the same and they are listed in Table 2. Top-of-atmosphere reflectances include the mean values and standard deviations of the native MODIS pixels inside the 10 km MODIS aerosol pixel. The optimal hyperparameter values found in the hyperparameter optimization are listed in Table 1.

### 3.3.2 Model enforced correction model

The model enforced correction approach takes the same set of input variables as the fully learned model together with some additional Dark Target related variables to predict the approximation errors for the AOD at 550 nm and AE. In the evaluation of the trained model enforced post-process correction model, an estimate of the approximation error $e(\mathbf{x})$ is first computed and Equation 1 is used to compute the corrected satellite AOD or AE as $\tilde{f}(\mathbf{x}) + \hat{e}(\mathbf{x})$ where $\hat{e}(\mathbf{x})$ is the machine learning based estimate of the approximation error $e(\mathbf{x})$. The input variables in the model enforced models for both AOD and AE are the same and they are listed in Table 2. Top-of-atmosphere reflectance inputs include the mean values and standard deviations of the native MODIS pixels inside the 10 km MODIS aerosol pixel. The wavelengths used for the Dark Target related variables are those that are delivered in the operational data product files. More information on the specific details of the Dark Target related variables can be found from the Algorithm Theoretical Basis Document (ATBD) (Levy et al., 2009). The optimal hyperparameter values found in the hyperparameter optimization are listed in Table 1.

## 4 Evaluation of the Models

To evaluate the model-derived aerosol data products, we first collocate the MODIS and AERONET observations. In the MODIS-AERONET collocation, we follow similar comparison protocol as in Petrenko et al. (2012). For collocated MODIS pixel and AERONET observation we require:

- The distance from the center of a MODIS pixel to an AERONET stations is less than 25 km.

- Each MODIS pixel corresponds to at least three AERONET observations within ±30 minutes from the satellite overpass.

We use the AERONET AOD at 500 nm and the AE 440–870 nm to compute the median AERONET AOD at 550 nm corresponding to the collocated satellite pixels. In the construction of the training dataset, all MODIS pixels fulfilling the above criteria are used. In the construction of the validation dataset, we compute spatial median values for the MODIS and temporal median values for the AERONET AOD and AE values corresponding to a single satellite overpass and fulfilling the above criteria. We use medians instead of averages as recommended in Sayer and Knobelspiesse (2019) to obtain more representative and outlier tolerant results.

To get realistic estimates for the accuracy of the models, validation is carried out with cross-validation. In our two-fold cross-validation, we randomly divide AERONET stations into two groups and use other group of AERONET stations for training and other group for validation. To take full advantage of the data and get a global estimate for the accuracies of the models, the training and testing are carried out two times with both combinations of the two groups. The AERONET stations and their groups used in this study are shown in Figure 2. We also considered to use a conventional cross-validation in which the full





dataset would have been randomly divided into training and validation groups as it is done for example in Lary et al. (2009).
In the conventional cross-validation approach, however, the validation dataset pixels would have almost always contained

remarkably similar pixels in the training dataset corresponding to the same AERONET station (e.g. two adjacent pixels). This
would have resulted into overly optimistic results. We tested the conventional cross-validation approach and obtained almost
perfect retrievals with the coefficient of determination $R^2 = 0.99$ for both the fully learned and model enforced models. This
is a similar result as reported in Lary et al. (2009). However, to get realistic estimates for the model accuracies, we decided to
divide the training and validation datasets based on AERONET stations so that an AERONET station is only present either in

training or validation datasets not in both.

Our goal is to get globally applicable results and we use the coefficient of determination $R^2$ based on correlation coefficient,
root mean squared error (RMSE), and median bias as the metrics to compare the datasets. For AOD datasets we also compute
the ratio of samples that are inside the Dark Target over land EE envelope of $\pm(0.05 + 15\%)$. In addition to AOD and AE, we
will also evaluate the aerosol index (AI) that is defined as

$$AI = AOD \cdot AE.$$

AI has been considered as a better proxy for cloud condensation nuclei (CCN) than AOD (e.g. Gryspeerdt et al., 2017), since
AI is more sensitive than AOD to the accumulation mode aerosol concentration. However, as the AE has not been reliable over
the land regions, the AI over land has not been properly usable either in satellite-based studies of aerosol cloud interactions.

## 5 Results

The developed model enforced post-process correction method was tested with MODIS Aqua satellite data over land. We
compare the post-process corrected datasets to the operational Dark Target over land data and to a conventional fully learned
machine learning retrievals. The number of samples and AERONET stations in training and validation datasets are shown in
Table 3.

Figure 3 shows the MODIS-AERONET AOD comparison for the MODIS Dark Target, fully learned regression-based

MODIS data, and model enforced post-process corrected Dark Target AOD. The model enforced correction model is clearly
the most accurate dataset measured with the samples inside the Dark Target EE envelope (85%), $R^2 = 0.87$, and RMSE=0.08.
With both of machine learning based datasets, similar median AOD bias of about 0.01 is obtained while in the MODIS Dark
Target, the median AOD bias is 0.02. The MODIS Dark Target data also shows the non-physical negative AOD retrievals
whereas the fully learned and model enforced machine learning based datasets do not have samples with negative AOD. In the

fully learned model with a RF regressor that cannot extrapolate values outside the training set, the non-negativity of retrieved
AOD was expected as AERONET AOD is always non-negative. As the model enforced correction model, however, uses the
MODIS Dark Target AOD as a starting point, this is a very good result as the model regardless of the negative Dark Target
AOD values learns to predict only non-negative AOD values.

Figure 4 shows the AE validation results. The results show that the satellite-based AE over land is clearly less accurate

quantity than the AOD. MODIS Dark Target is clearly the worst performing retrieval with a low information content and high





uncertainty. Furthermore, the Dark Target AE values are mostly concentrated around three different values. This is a clear indication of relatively poor performance of the Dark Target mixing of fine and coarse aerosol models over land which is the reason why there is no operational Dark Target AE product Mielonen et al. (2011). Both of the machine learning based models result in a relatively similar performance. The model enforced post-process correction has the best performance in all

the metrics we use.

Figure 5 shows the results for the AI datasets. The accuracy of the AI datasets is generally similar to AOD datasets. Measured with $R^2$ and RMSE the post-process corrected dataset has the best accuracy. The machine learning based models have median bias of 0.02 and the MODIS Dark Target is free of bias. The MODIS Dark Target AOD has positive bias and AE negative bias thus resulting into bias-free AI.

Figure 6 shows the AOD and AE error distributions for each dataset for four different AOD and AE ranges. For AOD, the model enforced post-process corrected model is clearly the best performing model for AOD < 0.5. For AOD larger than 0.5, the machine learning based models have negative bias but the range of error values is clearly smaller than in the Dark Target. The samples with AOD > 0.5, however, represent only about 5% of all data samples so more data is needed for more accurate assessment of the accuracy of the models with large AOD. For AE, AE > 1.0 that corresponds to fine particles clearly results

in smaller bias for machine learning based datasets than AE < 1.0. Generally, the error distribution is certainly narrower with machine learning based datasets than with Dark Target. With AE < 1.0, Dark Target results in smaller median bias than the other datasets.

We also evaluate our datasets by comparing the AOD and AE with a grouping based on the dominant aerosol types of the AERONET stations (Sogacheva et al., 2020). Figure 7 shows the error distributions for AOD and AE for background aerosol,

fine aerosol, and coarse aerosol dominated AERONET stations. The seasonal classification from Sogacheva et al. (2020) is used for the classification of the AERONET stations according to prevailing aerosol type. The results for both AOD and AE show that for background aerosol and fine dominated aerosol stations the machine learning based datasets clearly perform better than the Dark Target. The background aerosol dominated data forms the clear majority of the data (80 % of all samples) and the model enforced post-process corrected dataset is clearly the best performing dataset for these AERONET stations. For

coarse aerosol dominated AERONET stations, the Dark Target has smaller bias in the data than other datasets. Coarse aerosol dominated AERONET station data, however, has only about 3 % of all the samples both in training and validation and thus more data is needed for further assessment of the results with coarse aerosol data and for better training of the machine learning methods.

These results show that it is highly beneficial to combine both the physics-based retrieval algorithm and machine learning

based post-process correction.

## 6   Conclusions

A model enforced post-process correction method for the satellite aerosol retrievals was developed. In the correction method, a machine learning based model is trained to predict the approximation error in the conventional aerosol retrievals and the





estimate of the error is used to correct the retrievals. The proposed post-process correction approach is computationally efficient
and processing of the existing satellite aerosol datasets does not require the much larger radiance datasets. The proposed
approach is also generic in the sense that it does not require modifications to the original retrieval algorithm. The approach
is also flexible with respect to the machine learning model (e.g. neural network, Random Forest) which can be chosen case
specifically for each satellite dataset.

We found that the post-processing correction method resulted into significantly improved accuracy of the MODIS AOD and
AE retrievals over land. With the proposed correction we obtained AOD bias smaller than the accuracy of the accurate ground-
based AERONET AOD. Furthermore, the correction approach resulted in better accuracy retrievals than the conventional fully
learned machine learning based models in which the satellite observations are used to directly predict the accurate retrievals. In
many applications, even a small improvement in the aerosol characterization accuracy and precision could be translated into a
significant gain, e.g. in remote sensing of land surface derived products such as solar induced fluorescence or surface reflectance
based indices within the visible and near-infrared parts of the spectrum. Compared to the Dark Target algorithm performance,
the improved AOD and AE retrievals derived from the post-processing correction lead to a significant gain in the computation
of the Aerosol Index (AI) over land. The accurate AI retrievals, especially for the small AI values, are highly important for
example for the aerosol-cloud-interaction studies in which AI is commonly used as a proxy for the CCN concentration. Here
we observe that with the machine learning based retrievals there are significantly lower number of highly biased AI retrievals
especially corresponding to small AI values. Improvement of small AI retrievals are highly beneficial especially for the aerosol-
cloud-interaction studies. For the land satellite remote sensing community, any improvement in the aerosol characterization is
translated into an important gain in terms of the achieved satellite-derived surface reflectance accuracy. In this regard, the use
of the post-processing approach opens up the possibility to easily re-correct the long satellite-based land surface property time
series.

We also found that the conventional cross-validation, in which the pixels of the full dataset are randomly divided into training
and validation datasets, may lead to overly optimistic results in machine learning based algorithms for satellite retrievals. This
is because too similar pixels corresponding to the same AERONET station are in both training and validation datasets. In our
study, if we carried out this conventional cross-validation we would have obtained almost perfect retrievals with coefficient of
determination $R^2 = 0.99$. This is a similar result that can be found in some publications evaluating machine learning based
approaches for satellite retrievals. We tackled the cross-validation issue by dividing the data into training and validation datasets
by AERONET station.

Even though we tested the proposed approach with satellite aerosol data, our approach is not limited to aerosols only and
is generally applicable to different types of satellite data products as long as suitable training data is available. In addition to
observational data, simulated data could be suitable for training the post-process correction models in some applications. As
we use an ensemble method Random Forest for the correction, it could be possible to use the spread of the ensemble members
outputs to obtain pixel-based uncertainty estimates for the corrected retrievals. Furthermore, a sensitivity study for the post-
process correction models could provide us valuable information on the weak parts of the conventional retrieval algorithms
and they could be used as a tool to assess the retrieval sensitivity.



*Code and data availability.* The code and data to reproduce the results is available from the corresponding author on request.

*Author contributions.* AL and VK developed the methodology presented. AL collected and processed the data. AL, VK, PK, AK, TM, NS, LS, THV, and AA analysed the results. AL wrote the original manuscript. VK, PK, AK, TM, NS, LS, THV, and AA reviewed and edited the manuscript.

*Competing interests.* The authors declare that they have no competing interests.

*Acknowledgements.* We thank the AERONET PIs and their staff for establishing and maintaining the AERONET sites used in this investi-
gation. We thank NASA MODIS and Dark Target teams to kindly make the MODIS data publicly available. V. Kolehmainen acknowledges the Academy of Finland, Centre of Excellence in Inverse Modelling and Imaging (project 312343). The authors wish to acknowledge CSC – IT Center for Science, Finland, for computational resources.



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





**Table 1.** Hyperparameter values used for training the RF models.

| Hyperparameter | AOD Fully Learned | AOD Model Enforced | AE Fully Learned | AE Model Enforced |
|---|---|---|---|---|
| Number of trees | 360 | 320 | 400 | 360 |
| Maximum depth of a tree | 47 | 47 | 46 | 52 |
| Maximum number of features in a split | 68 % | 44 % | 84 % | 68 % |

**Table 2.** Input parameters of the fully learned and model enforced post-process correction aerosol retrieval models.

| Fully learned retrieval model | Model enforced post-process correction model |
|---|---|
| Top-of-atmosphere reflectances at 470, 550, 650, 860, 1240, 1630, and 2110 nm. | Top-of-atmosphere reflectances at 470, 550, 650, 860, 1240, 1630, and 2110 nm. |
| Sensor zenith and azimuth angles | Sensor zenith and azimuth angles |
| Solar zenith and azimuth angles | Solar zenith and azimuth angles |
| Scattering angle | Scattering angle |
| Land topographic altitude | Land topographic altitude |
| | Dark Target retrieved surface reflectances at 470, 660, and 2130 nm |
| | Dark Target retrieved AOD at 470, 550, and 660 nm |
| | Dark Target Angstrom exponent based on AOD retrieved at 470, 550, and 660 nm |
| | Dark Target retrieval quality flag |
| | Dark Target fine aerosol model used for land retrieval |

**Table 3.** Number of samples and AERONET stations in training and validation dataset groups.

| | | Group 1 | Group 2 | Total |
|---|---|---|---|---|
| **Training** | Number of data samples | 1 488 482 | 1 638 409 | 3 126 891 |
| | Number of AERONET stations | 278 | 277 | 555 |
| **Validation** | Number of data samples | 45 365 | 49 253 | 94 618 |
| | Number of AERONET stations | 262 | 265 | 527 |





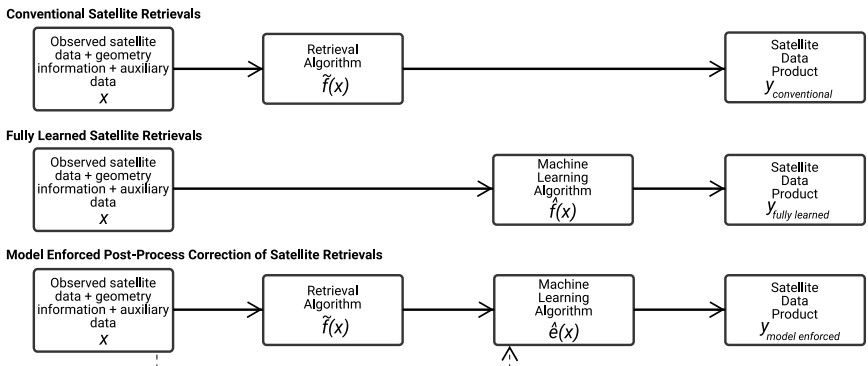

**Figure 1.** Top: Conventional satellite retrieval. Middle: Fully learned machine learning based satellite retrieval approach. Bottom: Model enforced post-process correction satellite retrieval approach.

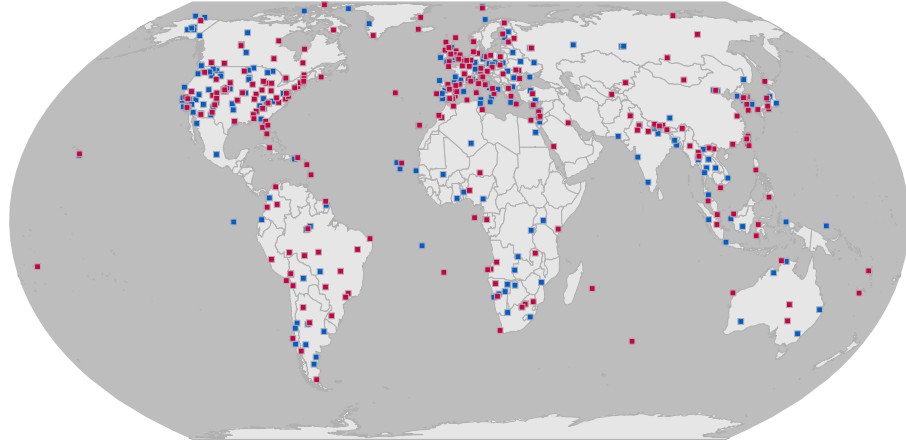

**Figure 2.** Locations of AERONET stations used in training and testing of the models. Red and blue colors indicate the random grouping of the stations used in the cross-validation.





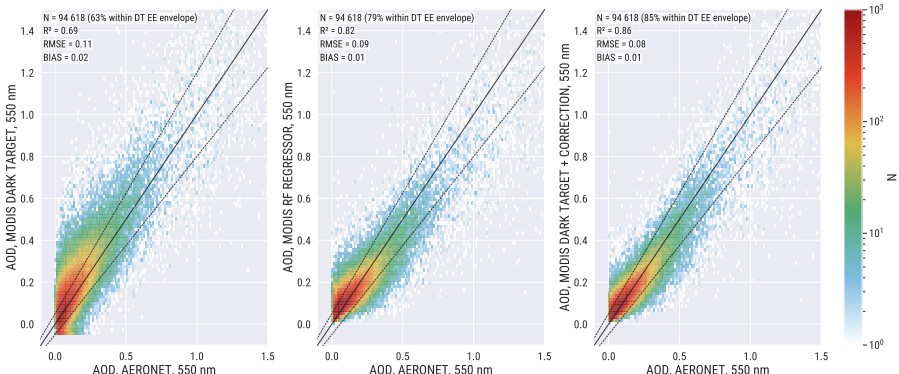

**Figure 3.** Comparison of AERONET and MODIS AOD at 550 nm. Left: MODIS Dark Target over land. Middle: MODIS fully learned Random Forest (RF) based regression model. Right: MODIS Dark Target with RF regression based model enforced post-process correction. The solid black line indicates the 1:1 line. The dashed black lines show the MODIS Dark Target expected error (EE) envelope.

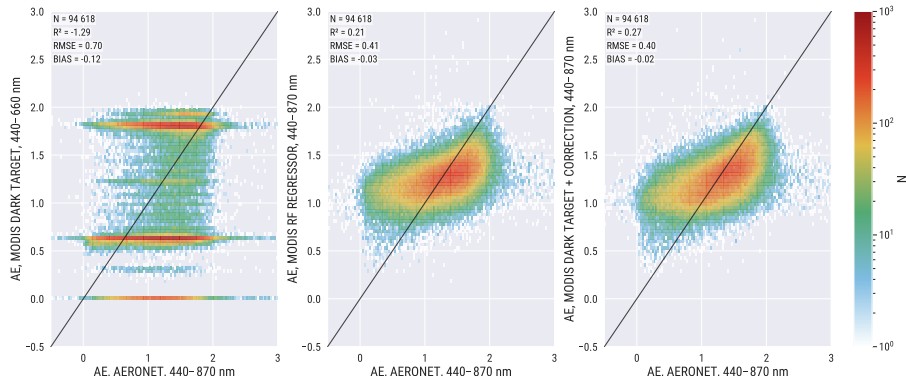

**Figure 4.** Comparison of AERONET and MODIS Angstrom exponent (AE). Left: MODIS Dark Target over land AE 440–660 nm. Middle: MODIS fully learned Random Forest (RF) based regression model AE 440–870 nm. Right: MODIS Dark Target AE 440–660 nm with RF regression based model enforced post-process correction. The solid black line indicates the 1:1 line.





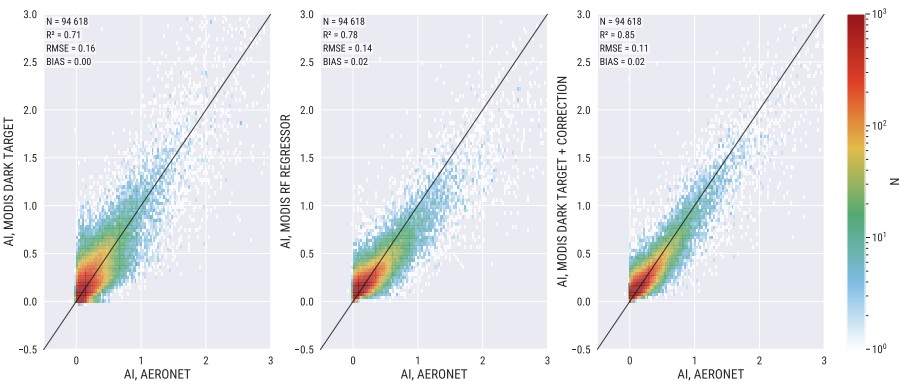

**Figure 5.** Comparison of AERONET and MODIS aerosol index (AI). Left: MODIS Dark Target over land. Middle: MODIS fully learned Random Forest (RF) based regression model. Right: MODIS Dark Target with RF regression based model enforced post-process correction. The solid black line indicates the 1:1 line.

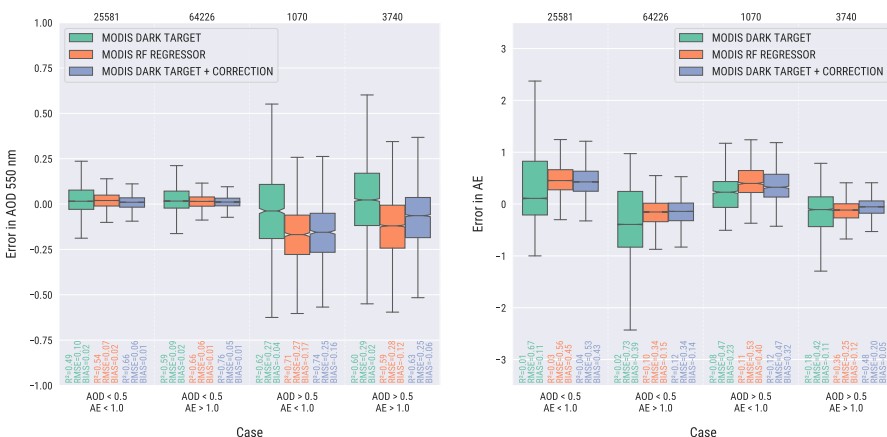

**Figure 6.** Error distribution in AOD at 550 nm (left) and Angstrom exponent (AE) (right) in different validation datasets. The data are grouped to four different groups based on AERONET AOD and AE. The numbers at the top of the figure indicate the number of validation samples in each group. The box shows the 25–75% quartiles of the datasets. The whiskers extend to display the rest of the distribution, except for points that are determined to be outliers. The notch in the box shows the 95% confidence interval around the median.

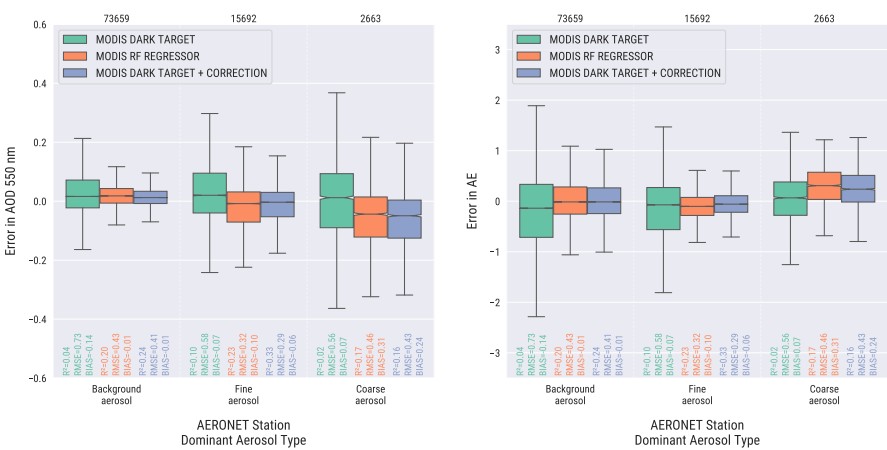

**Figure 7.** Error distribution in AOD at 550 nm (left) and Angstrom exponent (AE) (right) in different validation datasets. The data are grouped based on dominant aerosol type of AERONET stations based on Sogacheva et al. (2020). The numbers at the top of the figure indicate the number of validation samples in each group. The box shows the 25–75% quartiles of the datasets. The whiskers extend to display the rest of the distribution, except for points that are determined to be outliers. The notch in the box shows the 95% confidence interval around the median.