# Peer review of "Model Enforced Post-Process Correction of Satellite Aerosol Retrievals"

_Atmospheric Measurement Techniques, 2020_

## Referee Comment (RC1) · Anonymous Referee #1 · 20 Oct 2020

This work provides solid scientific knowledge, and the general quality is very good. It presents an interesting technique to obtain improved aerosol products based on satellite retrievals.

It could be interesting to see if there is a pattern found in the stations with better improvements, compared to those with fewer improvements (region / urban sites / etc.).

Did you find both AOD and AE large improvements at the same stations?

---

## Referee Comment (RC2) · Anonymous Referee #2 · 4 Jan 2021

Lipponen et al. have tried to use a model enforced post-process correction of satellite retrievals, especially the post-process correction of approximation error in the retrieval. It seems helpful as it allows to use other ancillary dataset and efficiently reprocess the existing aerosol dataset without much change in raw retrievals. I found the method innovative and potential for broader applications. However, the model has some unexplained characteristic that needs to be addressed.

Specific comments:

1. I found the abstract too general with having description only on the description/uncertainty and processing of the dataset by the algorithm. It lacks how much the AOD/AE retrievals were improved against a standard algorithm and any other specific observation (like regional or temporal advantages/ performance).

2. Abstract: modify line 1, as all satellite do not provide global data. Atmospheric characterization by aerosol does not mean anything.

3. Introduction: authors have used the word 'anthropogenic' too general, as it would be crude to decide the anthropogenic nature of aerosol based solely on high AE, as it could also be attributed by natural forest fire and other. Kindly revise the text.

4. Line 30: AE is not a typical satellite retrieval data as it is negative slope of AOD for a particular wavelength in log. Avoid writing AE as retrievals.

5. Section 3.1: what quality flag was considered to retrieve DT AOD to avoid cloud contamination and other errors in the AOD retrieval?

6. Line 195: Why authors have pointed that AE (and the AI) is not reliable over land?

7. As in Fig. 3, model-enforced correction model has achieved higher accuracy and lower bias compared to other. How the model-enforced processing are dependent on the selection of ancillary data? Is selection of different variable or a different group of variables have any possibility to change the outcome?

8. Line 215: I agree, due to poor mixing of aerosol models in DT over land there is no operational DT AE product. In Fig. 4, there is improvement in AE prediction by model-enforced correction but result is comparable to fully learned Random Forest based regression model. Why author have not tried FMF which is a quantitative value and provide better estimation of aerosol size? Selection of a different parameter would have leads to add the applicability of the result.

9. Clearly in Fig. 6, model-process data is well comparable to that of MODIS RF REGRESSOR for AOD<0.5 while at high AOD, machine learning based models have negative bias but the range of error values is clearly smaller. So, can we conclude from it that the post-processing of the AOD will not work well for the cases with extreme aod observation? How post-processing will improve aod estimation for a region with high AOD, like in China/ India?

---

## Author Comment (AC1) · 29 Jan 2021

The comment was uploaded in the form of a supplement:
https://amt.copernicus.org/preprints/amt-2020-229/amt-2020-229-AC1-supplement.pdf
* * *

---

## Author Comment (AC2) · 29 Jan 2021

We would like to thank the reviewers for their comments on this study. Below, we address the comments by the reviewers. The reviewers' comments are typed in bold and our replies to them in regular font. To help the reviewers, we also list some parts of the revised manuscript in our replies and these parts are typed in italic font or with quotation marks for small comments. Below our replies, a revised version of the manuscript showing all the changes made in the revision.

**1   Comments by the reviewer #1**

**This work provides solid scientific knowledge, and the general quality is very good. It presents an interesting technique to obtain improved aerosol products based on satellite retrievals.**

We are glad to hear you find our work interesting. Thank you for this encouraging feedback.

**It could be interesting to see if there is a pattern found in the stations with better improvements, compared to those with fewer improvements (region / urban sites / etc.).**

We carefully compared and analysed the seasonal average retrieval errors, as mean absolute AOD error, by station. In the analysis, we compared the ratio of the average errors between Dark Target and post-process corrected and compared this ratio, for example, to the measurement geometry related variables, AERONET AOD, AERONET station location elevation, and the amount of training data samples used for the post-process correction model. In these comparisons, we did not find a clear pattern in the AOD correction to explain the rate of improvements between different stations.

To include this information described above into the manuscript, the end of the results section now reads:

*To analyse the possible reasons for the improvements in the AOD accuracy, we also examined the correlations between the improvement in the AOD accuracy and some post-process correction model inputs and training parameters. In this analysis, we used seasonal averages of the improvements in the mean absolute error of AOD from each AERONET station. The input variables used for the AOD improvement correlation test included the measurement geometry related variables such as solar zenith angle, sensor zenith angle, and relative azimuth angle, AERONET AOD, AERONET station elevation, Dark Target fine aerosol model, and the amount of training data samples used to train the post-process correction model. We also analysed the geographic locations of the AERONET stations for which the post-process correction model performed best and worst by visual inspection. In these correlation analyses, we did not find a clear pattern to explain AOD accuracy improvements.*

*The results of the post-process corrected aerosol dataset show that it is highly beneficial to combine both the physics-based retrieval algorithm and machine learning based post-process correction.*

**Did you find both AOD and AE large improvements at the same stations?**

There is no clear correlation between the improvements in AOD and AE. We did not find the largest improvements in both AOD and AE at the same stations.

**2    Comments by the reviewer #2**

**Lipponen et al. have tried to use a model enforced post-process correction of satellite retrievals, especially the post-process correction of approximation error in the retrieval. It seems helpful as it allows to use other ancillary dataset and efficiently reprocess the existing aerosol dataset without much change in raw retrievals. I found the method innovative and potential for broader applications. However, the model has some unexplained characteristic that needs to be addressed.**

Thank you very much for your comments, we are happy to hear you think the method is innovative and there is potential for broader applications.

**Specific comments:**
**1. I found the abstract too general with having description only on the description/uncertainty and processing of the dataset by the algorithm. It lacks how much the AOD/AE retrievals were improved against a standard algorithm and any other specific observation (like regional or temporal advantages/ performance).**

The abstract was revised for better grammar and easier reading, and we also briefly list details on improvement of AOD retrievals. As the journal instructions advice the abstract to be short we do not want to add too much details in the abstract. The revised abstract is now of the form:

*Satellite-based aerosol retrievals provide a timely view of atmospheric aerosol properties, having a crucial role in the subsequent estimation of air quality indicators, atmospherically corrected satellite data products, and climate applications. However, current aerosol data products based on satellite data often have relatively large biases compared to accurate ground-based measurements and distinct uncertainty levels associated with them. These biases and uncertainties are often caused by oversimplified assumptions and approximations used*

*in the retrieval algorithms due to unknown surface reflectance or fixed aerosol models. Moreover, the retrieval algorithms do not usually take advantage of all the possible observational data collected by the satellite instruments and may, for example, leave some spectral bands unused. The improvement and the re-processing of the past and current operational satellite data retrieval algorithms would become tedious and computationally expensive. To overcome this burden, we have developed a model enforced post-process correction approach to correct the existing operational satellite aerosol data products. Our approach combines the existing satellite aerosol retrievals, and a post-processing step carried out with a machine learning-based correction model for the approximation error in the retrieval. The developed approach allows for the utilization of auxiliary data sources, such as meteorological information, or additional observations such as spectral bands unused by the original retrieval algorithm. The post-process correction model can learn to correct for the biases and uncertainties in the original retrieval algorithms. As the correction is carried out as a post-processing step, it allows for computationally efficient re-processing of existing satellite aerosol datasets without fully re-processing the much larger original radiance data. We demonstrate with over land aerosol optical depth (AOD) and Angstrom exponent (AE) data from the Moderate Imaging Spectroradiometer (MODIS) of Aqua satellite that our approach can significantly improve the accuracy of the satellite aerosol data products and reduce the associated uncertainties. For instance, in our evaluation, the number of AOD samples within the MODIS Dark Target expected error envelope increased from 63 % to 85 % when the post-process correction was applied. In addition to method description and accuracy results, we also give recommendations for validating machine learning-based satellite data products.*

**2. Abstract: modify line 1, as all satellite do not provide global data. Atmospheric characterization by aerosol does not mean anything.**

We have removed the word "global" and "atmospheric characterization by aerosol" in the revision. See also reply to the comment 1.

**3. Introduction: authors have used the word 'anthropogenic' too general, as it would be crude to decide the anthropogenic nature of aerosol based solely on high AE, as it could also be attributed by natural forest fire and other. Kindly revise the text.**

We have revised the text and the first paragraph of the introduction now reads:
*Climate change is one of the most serious problems humankind is facing today. Despite the long and active research, the future climate projections still contain significant uncertainties, and anthropogenic aerosol forcing currently comprises the largest source of this uncertainty (Pachauri et al., 2014). More accurate*

*information about the anthropogenic aerosols would help us improve our understanding of anthropogenic aerosol forcing, leading to a significant reduction of the uncertainties in future climate projections. Kaufman et al. (2005) and Yu et al. (2009) developed an algorithm to estimate anthropogenic aerosol component utilizing MODIS fine-mode aerosol optical depth (AOD) fraction and corrections to exclude fine mode natural dust and marine aerosols. The algorithm was only applicable over oceans due to the low accuracy of fine mode fraction over land, directly linked to inaccurate Angstrom exponent (AE). Therefore, any prospects to improve the satellite-based AE, particularly over land, could bring fundamental advances in measurement-based estimates of the global aerosol anthropogenic fraction. Another major global problem is air quality. In 2017, 2–25 % of all deaths globally were attributable to ambient particulate matter pollution (GBD 2017 Risk Factor Collaborators et al., 2018). To better monitor and understand air quality and pollution sources, near real-time global observations of aerosols are needed. In this respect, the only way to get wide coverage and near real-time information about atmospheric aerosols is to use satellite aerosol retrievals.*

**4. Line 30: AE is not a typical satellite retrieval data as it is negative slope of AOD for a particular wavelength in log. Avoid writing AE as retrievals.**

Revised according to the comment. Now the sentence on line 30 reads: "Satellite aerosol retrieval algorithms retrieve the aerosol optical properties such as AOD given the satellite observed top-of-atmosphere radiances or reflectances and the measurement geometry information. Other aerosol optical property-related quantities, such as the AE, are often derived from the retrieved aerosol optical properties." AE as an example of an accurate satellite retrieval was also removed from Section 2.

**5. Section 3.1: what quality flag was considered to retrieve DT AOD to avoid cloud contamination and other errors in the AOD retrieval?**

All quality flags were used in MODIS DT data. This selection was intentional as we also wanted to correct for the lower quality flagged data as well to improve the AOD retrievals at as many retrievals as possible. We have now mentioned this in the revised text by adding a sentence: "We accept all pixels with all quality flags in our datasets in this study.".

**6. Line 195: Why authors have pointed that AE (and the AI) is not reliable over land?**

The Dark Target team website says in their frequently asked questions: "Why can't I find fine mode AOD or Angstrom Exponent over land in collection 6? This product is not accurate over land and the MODIS aerosol group decided not to include it in collection 6." We have added a citation to a paper

(Levy et al., 2010) that concludes "MODIS does not provide quantitative information about aerosol size over land. Thus, we strongly recommend that users NOT use size products quantitatively."

**7. As in Fig. 3, model-enforced correction model has achieved higher accuracy and lower bias compared to other. How the model-enforced processing are dependent on the selection of ancillary data? Is selection of different variable or a different group of variables have any possibility to change the outcome?**

The accuracy of a correction model naturally depends on the information content of the inputs. We have analysed the correlations between the model inputs and the improvements in retrieval accuracy (see reply to Referee #1). Our analysis did not show any clear pattern or correlations between the inputs and improvement in the retrieval accuracy. This could indicate that the reason for the improvement varies from region to region or there may be multiple inputs together that are the major contributor to the accuracy improvement.

Machine learning, especially with the Random Forest regressor, however, is quite tolerant against having non-informative inputs in the models. In the possible case some of the input variables did not have information content to improve the correction model, the training of the Random Forest model would effectively leave these variables mostly unused. Therefore having too many input variable is not a big issue here and the model-enforced processing is not too dependent on the selection of ancillary data as long as there is enough information content in the inputs. Adding new inputs with additional, new information content, on the other hand, should improve the results even more.

**8. Line 215: I agree, due to poor mixing of aerosol models in DT over land there is no operational DT AE product. In Fig. 4, there is improvement in AE prediction by model-enforced correction but result is comparable to fully learned Random Forest based regression model. Why author have not tried FMF which is a quantitative value and provide better estimation of aerosol size? Selection of a different parameter would have leads to add the applicability of the result.**

Use of fine mode fraction would have lead to issues related on how to the fine mode fraction definition. In AERONET alone, different data products define the fine mode fraction differently. Whereas in the Dark Target retrieval algorithm, the FMF is defined as the ratio of fine aerosol model caused top-of-atmosphere reflectance to the total top-of-atmosphere reflectance (reflectance vs. aerosol loading). The use of FMF would therefore easily lead to "apples to oranges" comparison that we wanted to avoid. As AE is at least a qualitative indicator of aerosol size distribution but better defined, we decided to use it in our study. We thank the reviewer for pointing out this type of comparison and think that the comparison of FMF is a good topic for the future studies.

**9. Clearly in Fig. 6, model-process data is well comparable to that of MODIS RFREGRESSOR for AOD¡0.5 while at high AOD, machine learning based models have negative bias but the range of error values is clearly smaller. So, can we conclude from it that the post-processing of the AOD will not work well for the cases with extreme aod observation? How post-processing will improve aod estimation for a region with high AOD, like in China/ India?**

The number of samples corresponding to high AOD cases is clearly small compared to number of cases with small AOD. Therefore, more data is needed and it is not possible to conclude with this data that the post-processing of the AOD will not work well for the cases with extreme AODs - on the other hand, with these data, we cannot conlude either that the correction is working well with extreme AODs. More careful analysis of the method with high AODs is a good topic for future studies and clearly more data with extreme AODs is needed both for training the correction models and validating the results. More data exists already (full MODIS timeseries from both of the satellites could be used) so this would be a feasible but somewhat labourous task to carry out and we are considering to run post-process correction for full timeseries of data in the future.

We have already mentioned this issue in the manuscript's Results section: "*The samples with AOD > 0.5, however, represent only about 5% of all data samples so more data is needed for more accurate assessment of the accuracy of the models with large AOD.*" Now based on this referee comment we have revised the manuscript and added the following sentence to the conclusions and divided a paragraph into two: "*However, due to the small number of cases available with high AOD values, further studies would be required to better assess the post-process correction method's accuracy in high AOD scenarios.*"

Referee #2 had also two comments regarding our manuscript given in the earlier phase of the review. Below are the replies to these two comments.

**1. Please explain why DT 10 km data was used to compare.**

[revised manuscript text omitted]

---

## Referee Comment (RC3) · Anonymous Referee #1 · 11 Feb 2021

This work provides solid scientific knowledge, and the overall quality is very good. It presents an interesting technique to obtain improved aerosol products based on satellite retrievals.

All the questions posed by the reviewers have been properly addressed, therefore it should be accepted in the current form.
* * *

---

## Author Comment (AC3) · 12 Feb 2021

We would like to thank the reviewer for the comments.

**1 Comments by the reviewer #1**

**This work provides solid scientific knowledge, and the overall quality is very good. It presents an interesting technique to obtain improved aerosol products based on satellite retrievals.**

**All the questions posed by the reviewers have been properly addressed, therefore itshould be accepted in the current form.**

We are happy to hear you find our work very good in quality. Thank you for your interest and time used for reading the manuscript.